# Iron Release Profile of Silica-Modified Zero-Valent Iron NPs and Their Implication in Cancer Therapy

**DOI:** 10.3390/ijms20184336

**Published:** 2019-09-04

**Authors:** Li-Xing Yang, Ya-Na Wu, Pei-Wen Wang, Wu-Chou Su, Dar-Bin Shieh

**Affiliations:** 1Institute of Basic Medical Sciences, National Cheng Kung University, Tainan 70101, Taiwan; 2Institute of Oral Medicine and Department of Stomatology, College of Medicine, National Cheng Kung University Hospital, National Cheng Kung University, Tainan 70101, Taiwan; 3Institute of Biological Chemistry, Academia Sinica, Taipei 11529, Taiwan; 4Department of Internal Medicine, Division of Hematology/Oncology, National Cheng Kung University Hospital, College of Medicine, National Cheng Kung University, Tainan 70101, Taiwan; 5Center of Applied Nanomedicine, National Cheng Kung University, Tainan 70101, Taiwan; 6Center for Micro/Nano Science and Technology, Advanced Optoelectronic Technology Center, Innovation Center for Advanced Medical Device Technology, National Cheng Kung University, Tainan 70101, Taiwan

**Keywords:** zero-valent iron, silica, reactive oxygen species, lysosome, iron release

## Abstract

To evaluate the iron ion release profile of zero-valent iron (ZVI)-based nanoparticles (NPs) and their relationship with lysosomes in cancer cells, silica and mesoporous silica-coated ZVI NPs (denoted as ZVI@SiO_2_ and ZVI@mSiO_2_) were synthesized and characterized for the following study of cytotoxicity, intracellular iron ion release, and their underlying mechanisms. ZVI@mSiO_2_ NPs showed higher cytotoxicity than ZVI@SiO_2_ NPs in the OEC-M1 oral cancer cell line. In addition, internalized ZVI@mSiO_2_ NPs deformed into hollow and void structures within the cells after a 24-h treatment, but ZVI@SiO_2_ NPs remained intact after internalization. The intracellular iron ion release profile was also accordant with the structural deformation of ZVI@mSiO_2_ NPs. Burst iron ion release occurred in ZVI@mSiO_2_-treated cells within an hour with increased lysosome membrane permeability, which induced massive reactive oxygen species generation followed by necrotic and apoptotic cell death. Furthermore, inhibition of endosome–lysosome system acidification successfully compromised burst iron ion release, thereby reversing the cell fate. An in vivo test also showed a promising anticancer effect of ZVI@mSiO_2_ NPs without significant weight loss. In conclusion, we demonstrated the anticancer property of ZVI@mSiO_2_ NPs as well as the iron ion release profile in time course within cells, which is highly associated with the surface coating of ZVI NPs and lysosomal acidification.

## 1. Introduction

Based on reports from GLOBOCAN, cancer burden had risen to 18.1 million new cases and 9.6 million cancer deaths worldwide in 2018 [1]. Therefore, improving the therapeutic efficacy and outcomes of cancer treatment is urgent. Innovative nanomedicine, which has emerged recently, has shown several promising advantages over conventional cancer therapies, including early detection, improved treatment efficacy, and early diagnosis of cancer. Some metal or metal oxide nanomaterials, including zinc oxide [2,3], iron-based core-shell nanoparticles (NPs) [4,5], zero-valent iron nanoparticles (ZVI NPs) [6], and some iron-containing metal complexes [7,8], have demonstrated outstanding cancer-selective cytotoxicity. Recent studies suggested the role of electrophilic theory in chemical carcinogenesis where free radicals derived lipid peroxidation may play certain roles. A QSAR mechanistic interpretation to bridge the mutagenesis and carcinogenesis has been proposed [9,10]. Moreover, such NPs have been reported to kill cancer cells through oxidation therapy. In oxidation therapy, reactive oxygen species (ROS) are generated selectively in cancer cells for killing them while sparing healthy cells. Iron oxide (Fe_3_O_4_) NPs also show the ability to selectively kill cancer cells through the induction of proapoptotic genes and tumor suppressor genes [11]. Some reports have shown that ROS can initiate necrosis and apoptosis in various cell types [12,13]. Furthermore, ROS production has been extensively observed for serial chemotherapeutic compounds, including vinblastine, cisplatin, and paclitaxel [14,15].

ZVI NPs have been commonly used for environmental decontamination over the past decades [16,17,18,19]. Generally, ZVIs are used as reductants, and they produce ROS, thereby promoting the Fenton reaction, through which they degrade various organic pollutants [20]. A typical Fenton reaction is mediated through the interaction of ferric, ferrous ions, and hydrogen peroxide to generate hydroxyl radicals, which are strong oxidants capable of decomposing various organic contaminants. ZVI NPs have been recently reported to be potential bactericidal [21,22] and anticancer agents [6] owing to their excellent ROS-inducing properties. However, the volatile reactivity of ZVI NPs is one of the major concerns for biomedical applications, which results in the vigorous oxidation and reduced magnetic susceptibility of the particles [23]. Therefore, passivated coating of ZVI NPs is necessary for some biomedical applications. Among the various surface coating materials, silica coating has attracted much attention owing to its low cost and excellent biocompatibility [24,25].

Some research reports from independent research groups have stated that iron-based NPs have demonstrated promising anticancer activity in the past decade. Xu et al. used iron-platinum NPs as an iron reservoir for the controlled release of iron to inhibit the growth of various tumors in vitro [7]. Ahmed M et al. demonstrated the selective killing effect of Fe_3_O_4_ NPs for cancer cells through ROS and the p53 pathway [11]. Iron/silicon/carbon complex NPs were synthesized and their cancer-selective activity was identified by a research team from the United States [26]. The anticancer properties of ZVI NPs were also identified in 2019 [27]. In our research, iron core/gold shell NPs demonstrated cancer-selective cytotoxicity in serial oral cancer cells in vitro and in vivo [4,5] as well as in colorectal cell lines in vitro [28]. We found that such cancer-selective cytotoxicity of ZVI-based NPs was achieved through ferroptosis induction, and the resensitization strategy was also proposed to treat ZVI-resistant cancer cells [6]. Because iron is one of the most abundant elements in the human body, one of the advantages of ZVI-based NPs is that they will ultimately oxidize and decompose into highly biocompatible products [29], which the human body can easily detoxify through the iron metabolizing and buffering system.

Despite promising reports from different research groups indicating that iron-based NPs exhibit excellent cancer-selective cytotoxicity and their potential application as novel chemotherapy agents, the actual mechanisms of the cancer-specific cytotoxicity of iron-based NPs require further investigation. It is believed that the cytotoxic mechanisms of metal-containing NPs are mediated through the lysosome-enhanced Trojan horse effect [30]. However, the actual iron ion release profile of ZVI-based NPs and their relationship with lysosomes in cancer cells require further investigation.

In this study, we used silica and mesoporous silica as the shell coating on ZVI NPs (denoted as ZVI@SiO_2_ and ZVI@mSiO_2_) for the iron release profile investigation. ZVI@SiO_2_ was designed to serve as a sealed shell that blocked iron release from the ZVI core and was compared with ZVI@mSiO_2_. Series-parallel comparisons of both NPs, including cytotoxicity, ROS production level, intracellular iron release profile, and intracellular particle conformational change, were conducted in the following study. In addition, the therapeutic efficacy and safety were also evaluated in vivo to characterize the anticancer ability of ZVI@mSiO_2_ NPs for determining their future medical application potential.

## 2. Results

### 2.1. Characterization of ZVI@SiO_2_ and ZVI@mSiO_2_ NPs

ZVI NPs fully covered with silica and mesoporous silica-coated ZVI NPs were synthesized according to the protocols described by Ta-I Yang et al. and Yu-Shen Lin et al. for iron releasing effect against cancer cells study [31,32]. The synthesized ZVI@SiO_2_ NPs under transmission electron microscopy (TEM) showed well-coated, smooth, and thin layer silica shells around the ZVI cores (Figure 1a), with an average diameter of 28.8 ± 7.1 nm. ZVI@SiO_2_ NPs showed a narrower size distribution and a smaller particle diameter than ZVI@mSiO_2_ NPs (Figure 1b). By contrast, ZVI@mSiO_2_ showed a rough and thicker mesoporous silica shell, with a size distribution of 44.7 ± 12.0 nm (Figure 1d,e). A similar elemental composition of ZVI@SiO_2_ and ZVI@mSiO_2_ NPs was observed through energy dispersive spectrometer (EDS) analysis (Figure 1c,f). Iron element was the major content of the NPs, and the silica and oxygen contents in these two NPs were relatively low.

### 2.2. Different Shells of ZVI NPs Showed Distinct Cytotoxicity and ROS Induction Profile in the OEC-M1 Oral Cancer Cell Line

OEC-M1 is an unique oral cancer cell derived from a patient who had betel nut chewing history, which is one of the major causes of oral cancer in South East Asia. According to our previous serial studies [4,5,6,26,27], OEC-M1 was applied in this study as the test model for assessing the anticancer potential of ZVI@SiO_2_ and ZVI@mSiO_2_ NPs. The 3-(4,5-dimethylthiazol-2-yl)-2,5-diphenyltetrazolium bromide (MTT)-based cytotoxic assay showed that ZVI@mSiO_2_ NPs with the porous silica shell were cytotoxic to OEC-M1 cells, whereas ZVI@SiO_2_ NPs with the sealed silica shell were significantly less toxic to the cells (Figure 2a, *P* = 0.0342). Both types of NPs were non-toxic to normal human oral keratinocyte cells (hNOK) (Appendix A). Furthermore, the intracellular ROS level was dramatically different between the cells treated with the two NPs. ZVI@SiO_2_ NPs only induced negligible ROS production in OEC-M1 cells even after 24-h treatment, whereas the cells treated with ZVI@mSiO_2_ NPs produced drastic ROS induction immediately after 1-h treatment; the ROS induction persisted up to 24 h (Figure 2b).

### 2.3. ZVI@mSiO_2_ NPs Showed Conformation Change within OEC-M1 Cells

We further explored the interaction of ZVI NPs within the cells to investigate the various significant reactions of OEC-M1 against the two shell types of ZVI NPs. OEC-M1 cells treated with 10 μg/mL of ZVI@SiO_2_ or ZVI@mSiO_2_ NPs, respectively, were collected and processed for TEM imaging. Figure 3 shows the dramatically different particle conformation of ZVI@SiO_2_ and ZVI@mSiO_2_ within cells. Most ZVI@SiO_2_ NPs within OEC-M1 cells remained intact with a solid structure (Figure 3a), whereas ZVI@mSiO_2_ NPs showed hollow and void structures after 24-h treatment (Figure 3b). In addition, the NPs purified from cells treated for 24 h showed a significant structural change in ZVI@mSiO_2_ that deformed into hollow spheres (Figure 3d), while the ZVI@SiO_2_ still maintained the original structure (Figure 3c).

This finding suggested that the fully covered silica shell provided better protection to ZVI cores than the mesoporous silica shell. In addition, the cores of ZVI@mSiO_2_ NPs were empty, which suggested that ZVI cores might erode within the cells. The conformational change of particles and the drastic ROS burst in treated cells were coincidently observed in ZVI@mSiO_2_-treated cells. Oxidative stress has been reported to be induced by zero-valent iron NPs and ferrous ions in human cells [33]. Thus, it is crucial to explore whether a part of the cores of NPs was converted into iron ions, thereby leading to the ROS burst observed after treatment.

### 2.4. Intracellular Iron Ion Burst Only Observed in ZVI@mSiO_2_ NP-treated Cells Through Lysosome Acidification

We collected NP-treated cells and separated soluble cytosol factions and NP-containing insoluble portions at different time points after treating the cells with 10 μg/mL ZVI@SiO_2_ or ZVI@mSiO_2_ NPs to investigate the intracellular iron release profiles of NP-treated cells. The iron ion concentration in different compartments was then measured using ICP-MS. The ZVI@SiO_2_^𢈒^treated group showed 0.4 μg iron ions per milligrams of total protein (μg/mg protein) within OEC-M1 cells, measured after 2 h, and then, the concentration of intracellular iron ions gradually decreased. The total intracellular iron concentration, including the released iron ions and insoluble particle-type iron, reached a plateau of 2.7 μg/mg protein after 2-h treatment (Figure 4a). By contrast, ZVI@mSiO_2_-treated cells showed rapidly accumulated intracellular iron ions up to 1.5 μg/mg protein in an hour, and the soluble iron ions increased up to 3.9 μg/mg protein after 24 h. In addition, the total intracellular iron concentration continued to increase to 7.9 μg/mg protein after 24-h treatment (Figure 4c). Overall, ZVI@mSiO_2_ showed approximately four times higher internalizing efficiency than ZVI@SiO_2_.

The percentile composition of soluble ionic and insoluble particle types of total intracellular iron was calculated to parallelly compare the ratio of ionic iron to insoluble iron (Figure 4b,d). ZVI@SiO_2_-treated cells showed that only approximately 20% of the total intracellular iron was in ionic form in the cytosol (Figure 4b), whereas ZVI@mSiO_2_-treated cells showed that over 70% of the intracellular iron was in the ionic form within cytosol within an hour then decreased to approximately 50% after 2-h treatment (Figure 4d). Overall, approximately 50%–70% of internalized ZVI@mSiO_2_ was present in the ionic form, the percentile of which was much higher than that of ZVI@SiO_2_.

ZVI has been reported to be sensitive to the acidic environment, which accelerates the degradation and oxidization of ZVI into ferric ions [7,34]. Notably, the lysosome is the most acidic abundant organelle in the cell, and it digests most of the extracellular materials absorbed by the cell [28]. Therefore, to evaluate whether the acidification of lysosome is the crucial factor in the release of ionic ions, OEC-M1 oral cancer cells were cotreated with ZVI@mSiO_2_ and 20 mM ammonium chloride (NH_4_Cl), which is an inhibitor of endosome–lysosome system acidification [35]. Figure 4e,f shows the dramatic inhibition of iron ion release of NPs, which decreased from 3.9 to 0.8 μg/mg protein, without affecting the total NP uptake (Figure 4e). The percentage of ionic iron in the NH_4_Cl-treated group significantly reduced to 22% (Figure 4f, P = 0.0003) compared with ZVI@mSiO_2_ NP-alone group. Also, we observed the lysosome membrane permeability (LMP) increased after 1-h treatment of ZVI@mSiO_2_ that may lead to iron ions escape from lysosome to the cytosolic space (Appendix A).

### 2.5. ZVI@mSiO_2_ Induced Necrosis and Apoptosis in Lysosome Acidification and ROS Dependent Manner

Despite few articles reporting ZVI-induced cytotoxicity in human cells [33,36], the detail molecular mechanisms were limited. In this study, as shown in Figure 5a, ZVI@mSiO_2_ induced cell death in approximately 70% OEC-M1 cells after 24-h treatment. To further investigate the types of cell death induced by the NPs treatment, the Annexin V-affinity assay were performed at time points of 1-, 4-, and 8-h (Appendix A). The double positive populations were dramatically increased at as early as the first hour, whereas minor increase of the Annexin V positive/propidium iodide (PI) negative population was observed after 4-h treatment. Furthermore, there is no observable increase in caspase 3 activity in ZVI@mSiO_2_ treated cells at 12- and 24-h treatment (Appendix A). Necrostatin-1 was found to rescue the cytotoxicity caused by ZVI@mSiO_2_ exposure in a dose-dependent manner (Figure 5c). Therefore, it is concluded that the majority of ZVI@mSiO_2_ treated cells conferred necrosis and only a minor population underwent apoptosis.

Co-treatment of the cells with either 20 mM NH_4_Cl (an endosome–lysosome acidification inhibitor) or 200 μM vitamin C (an ROS scavenger) was able to rescue the NPs induced necrotic and apoptotic cell death as revealed by flow cytometry analysis and the overall cell viability presented in MTT assay. With further combination of NH_4_Cl and vitamin C in ZVI@mSiO_2_ co-treatment, we observed further increase in the overall cell viability to 70% and reduction in the necrotic as well as apoptotic population to 19% (the last panel in Figure 5a). Furthermore, cell viability was also rescued by co-treatment with deferoxamine (DFO) (Figure 5b). 

### 2.6. ZVI@mSiO_2_ NPs Inhibited Tumor Growth without Body Weight Loss

Our previous study reported on a series of nonoxidized zero-valent iron-containing particles that demonstrated an exceptional anticancer effect in vitro and in vivo [4,5,6,28,29]. Therefore, to evaluate and characterize the anticancer property of ZVI@mSiO_2_ in vivo, tumor-bearing mice were intravenously administered a single dose of PBS or 1 mg of ZVI@mSiO_2_ NPs in a PBS suspension, respectively, and tumor growth and body weight were then monitored regularly. As shown in Figure 6, ZVI@mSiO_2_ treatment could significantly reduce the tumor growth rate (*P* = 0.0029, Figure 6a) and even cause tumor shrinkage. Furthermore, no significant body weight loss was noted in the two groups of mice (Figure 6b).

## 3. Discussion

The total amount of internalized ZVI@mSiO_2_ was much higher than ZVI@SiO_2_ (Figure 4), which probably resulted from the surface roughness [37]. Therefore, the percentile calculation was further applied in this study to evaluate iron ion leakage instead of the total iron uptake. Furthermore, one oral cancer cell line, OEC-M1, was used as a model cell line to investigate iron ion release; OEC-M1 is typically one of the ZVI-sensitive model cell lines used in our serial studies published in different quality peer-reviewed journals. Because this research focused on the lysosomal interaction of ZVI NPs within ZVI-sensitive cells, the OEC-M1 cell line was applied as the model in this study. Nonetheless, the interaction of ZVI-based NPs and lysosomes in ZVI-insensitive cells reported elsewhere [6] requires further research.

In this study, we directly measured the iron release profile within the cells instead of in a mimic buffer system, which has been applied in most studies [7,30]. The finding suggested that the burst release of iron ions was lysosome dependent, which provided direct evidence to fill in the current knowledge gap. Metallic ion release triggered by the low pH in lysosomes has been a topic of discussion for years, and most studies have revealed that the ion release process is time-dependent and occurs over days [30]. However, the rapid and massive intracellular ion release within hours of treatment with ZVI-based NPs had not been discussed before, which can be one of the reasons for the dramatic anticancer effect observed in the current in vivo study. ZVI@mSiO_2_ showed the immediate burst release of iron ions, whereas the amount of iron ions released in ZVI@SiO_2_-treated cells was relatively low. We found that ZVI@mSiO_2_ NPs could be converted mostly into iron ions within an hour. This release profile was different from that of the iron oxide NPs, suggesting that ZVI is much more sensitive to the intracellular acidic environment than iron oxide. It has been reported that intra-lysosomal iron would induce LMP in trabecular meshwork cells [38]. We also found ZVI@mSiO_2_ NPs were able to increase cancer cell LMP after 1-h treatment. This time point was also correlated to the intracellular ROS induction. The loss of lysosome membrane integrity may lead to iron ions escape from lysosome to the cytosolic space. In the cell death analysis, ZVI@mSiO_2_ induced significant increase in the annexin V/PI double positive stained cells after 1-h treatment that could be regarded as necrosis rather than late apoptosis. After 4-h treatment, the annexin V positive/PI negative population slightly increased, suggesting a small proportion of affected cells underwent apoptosis. Besides, the caspase activity assay showed no obvious caspase activation upon ZVI@mSiO_2_ NPs treatment, indicating potential roles of caspase-independent apoptosis in the process. Taken together, we propose that the internalized ZVI@mSiO_2_ NPs were rapidly converted into iron ions in the acidic lysosome after uptake by the cancer cells. It resulted in the enhanced LMP and subsequent ROS induction that conferred cancer cells undergoing necrotic cell death, while a minor population underwent apoptotic cell death. This hypothesis was further supported by the discovery that the iron chelator, lysosome activity inhibitor, antioxidant and the necrosis inhibitor were all capable of reversing cell cytotoxicity induced by the NPs treatment. These results confirmed the lysosome-dependent iron ion release and ROS mediated necrosis to be the central mechanism of ZVI@mSiO_2_-induced cytotoxicity.

In addition, we directly observed that the intracellular particle conformation changes after NP uptake by the cells. The structure of ZVI@mSiO_2_ NPs became hollow, indicating that the core ZVI part was degraded, but most of ZVI@SiO_2_ NPs showed solid ZVI core structures, thereby suggesting that the sealed silica shell can protect the particles from degradation. Most studies tracking the intracellular metallic NPs structure have primarily focused on heavy metals such as gold, silver, and platinum or on metal oxides such as iron oxide, zinc oxide, and titanic oxide; the conformational change of NPs within the cells was seldom discussed in these studies [39,40,41,42]. In the present study, the significant structural deformation of ZVI@mSiO_2_ within the cells into a void structure supported the concept of massive and rapid intracellular iron ion release.

In the human cancer bearing animal model, the ZVI@mSiO_2_ NPs can effectively suppress cancer growth after a single dose intravenous injection. This observation suggests that passive targeting to the tumor lesion by enhanced permeability and retention effect may play a certain role. According to Heneweer and Wilhelm, between 0.7% to 5% of total intravenously administered dose of NPs formulation may ultimately reach the tumor site [43,44]. As 1 mg NPs was introduced to experimental mice bearing 100 mm^3^ size of tumor, the calculated concentration of NPs delivered to the tumor site would be as above the level of 1 mg × 0.7% / 100 mm^3^ = 70 μg/mL. Such dose is within the therapeutic window of the ZVI@mSiO_2_ NPs to induce cancer cell death according to the in vitro study. 

In summary, the burst release of iron ions of ZVI@mSiO_2_ was well characterized, and it was identified to be a lysosomal dependent process (Scheme 1). The in vivo study showed that a single-dose administration of ZVI@mSiO_2_ significantly inhibited tumor growth without observable weight loss, thereby exhibiting its excellent potential as an anticancer agent. The mesoporous silica coating seems to have effectively increased the particle uptake. With the highly biocompatible properties of silica, such an observation could be applied in different areas as a tool to boost total particle uptake. With the magnetic property of ZVI@mSiO_2_, such nanomaterial design can be further developed as theragnostic agents for advanced cancer therapy.

## 4. Materials and Methods

### 4.1. Synthesis of ZVI@SiO_2_ and ZVI@mSiO_2_ NPs

For ZVI@SiO_2_ NP synthesis, 0.16 mM iron (III) chloride, 0.16 mM citric acid, and 0.08 mM oleic acid were dissolved in 190 mL deionized water with stirring for 30 min under an argon atmosphere. Ten milliliters of 0.6% NaBH_4_ solution was then added to the mixture and stirred for 20 min at room temperature to form ZVI core NPs (Solution A). Following the addition of 196 μL of tetraethyl orthosilicate (TEOS) and 14 μL of 3-(aminopropyl)triethoxysilane (APTES) and mixing for 3 h, ZVI NPs were coated with a thin layer of the silica shell. The derived NPs were repeatedly washed with ethanol to remove any residual contaminants and then vacuum dried for storage at room temperature [31]. For ZVI@mSiO_2_ NP synthesis, the initial synthesis protocols were the same till solution A was derived. Solution A was added with 0.4 g CTAB and stirred until the solution turned clear. TEOS (196 μL) and APS (14 μL) were added to form the mesoporous silica shell [32].

### 4.2. TEM Characterization and EDS Analysis

The as-synthesized NPs and the particles purified from OEC-M1 cells after 24-h treatment were characterized using a high-resolution analytical electron microscope (JEM-2100F Electron Microscope /JEOL Co. 200 KV) equipped with an energy dispersive X-ray spectrometer (EDS). Cells treated by 10 μg/mL NPs were processed by lysis buffer (M-PERTM mammalian protein extraction reagent, Thermo Fisher Scientific) after 24-h treatment. Then the lysates were centrifuged at 20,000× *g* for 20 min to separate the intracellular undigested NPs in the pellet. The purified NPs were then resuspended in ethanol. The ultrastructure, size, and size distribution as well as the elemental compositions were analyzed and recorded. Five microliters of each sample was diluted with absolute alcohol to a final concentration of 10 μg/mL and then applied onto copper grids. They were then vacuum dried on the grid before TEM observation.

### 4.3. Cell Culture

The oral squamous cell carcinoma cell line OEC-M1, derived from Taiwanese male patients with oral cancer with a history of betel nut chewing [45], was kindly provided by Dr. Kuo-Wei Chang (Institute of Oral Biology, National Yang-Ming University, Taiwan) and cultured in Roswell Park Memorial Institute medium (RPMI) 1640 supplemented with 10% FBS and 1X antibiotics (Gibco) in a humidified incubator at 37 °C under 5% CO_2_. hNOK (normal human oral keratinocytes) was isolated from healthy donor and was cultured in Keratinocyte-SFM supplemented EGF, bovine pituitary extract (Gibco) and 1X antibiotics (Gibco) at 37 °C under 5% CO_2_. The primary cells were obtained with permission from Institutional Review Board of the Cheng-Kung University Hospital (No. B-ER-104-125) and under informed consent of the donors.

### 4.4. Cell Fixation and Embedding for TEM Imaging

A total of 10 μg/mL of ZVI@SiO_2_- or ZVI@mSiO_2_-exposed OEC-M1 cells (1 × 10^6^) were grown in a 10-cm diameter dish for 24 h. The cells were then collected and fixed in 2.5% glutaraldehyde in phosphate buffer for 4 h at 4 °C. After washing with 0.1 M cacodylate, the cells were incubated in 1% osmium tetroxide and 1.5% potassium ferricyanide in 0.1 M cacodylate for 1 h at room temperature. Samples were then washed, followed by dehydration in a graded ethanol series of 70%, 80%, 90%, and 100%. The dehydrated samples were then infiltrated with Spurr-ethanol solutions containing 25%, 50%, 75%, and finally 100% resin (30 min per stage). Thereafter, those resin-infiltrated specimens were polymerized at 70 °C for 48 h. The resin blocks were cut using an ultratome (Ultracut S, Leica Reichart) into 70-nm ultrathin sections by using a diamond knife. The thin sections were then post-stained with uranyl acetate and lead citrate. Transmission electron microscopy was conducted on a JEOL JEM-1400 at 120 kV, and images were captured using a CCD camera (Erlangshen, Gatan, Pleasanton, CA).

### 4.5. Cell Viability Analysis

In this study, the MTT assay was used to assess cell viability. Notably, actively respiring enzymes of cells convert water-soluble MTT to insoluble purple formazan. Formazan is then solubilized, and its concentration is determined through optical density. OEC-M1 cells in the log phase were seeded at a density of 5000 cells per well in a 96-well culture plate to evaluate the cytotoxicity of the NPs. The cells then received the assigned treatment, which included different concentrations of the NPs with or without 200 μM deferoxamine (DFO), 20 mM NH_4_Cl, 200 μM vitamin C or 5, 10, 20 μM necrostatin-1. After 24 h of incubation, fresh complete medium containing 0.5 mg/mL MTT assay solution was applied to replace the original culture medium, and the cells were further incubated for an hour. After removing the medium, dimethyl sulfoxide was added to dissolve the cells, and the MTT crystal was quantified through optical density at 490 nm for analyzing cell viability (Sunrise absorbance microplate reader, Tecan).

### 4.6. Intracellular Iron Release Assay

OEC-M1 cells seeded at a density of 500,000 cells per well in 6-well plates were treated with 10 μg/mL ZVI@SiO_2_ or ZVI@mSiO_2_ NPs with/without 20 mM NH_4_Cl for 1,2,4,8, and 24 h. The cells with the assigned treatment were collected and treated with the lysis buffer (M-PER^TM^ mammalian protein extraction reagent, Thermo Fisher Scientific) to lyse all intracellular lipid layers. The lysed cells were centrifuged at 20,000×*g* for 20 min to separate the intracellular undigested NPs (in pellet part) from the cytosolic fractions containing the released iron ions (in the supernatant). The two parts of the samples were further treated with aqua regia overnight and then diluted 20-fold with ddH_2_O. The iron concentrations of all the samples were then analyzed using inductively coupled plasma mass spectrometry (ICP-MS). The results were then normalized with the total protein of each group measured using the protein quantification kit (Pierce 660 nm Protein Assay Reagent, Thermo Fisher Scientific).

### 4.7. Intracellular ROS Analysis

Intracellular ROS production was measured based on their reaction with 2′,7′-dichlorodihydrofluorescein diacetate (H_2_DCFDA) to form the fluorescent compound 20,70-dichlorofluorescein (DCF). The cells were seeded at a density of 150,000 cells per well in 6-well dishes overnight to allow attachment. The cells were then cultured with ZVI@SiO_2_ or ZVI@mSiO_2_ NPs for 1,8, and 24 h, and 10 μM H_2_DCFDA was added at the end of the time course, with further incubation for 20 min. The cells were then harvested and analyzed immediately. The production of DCF fluorescence was monitored using FACS Canto II (excitation wavelength, 488 nm; emission wavelength, 515–545 nm), and the results were analyzed using FACS DIVA software.

### 4.8. Apoptosis and Necrosis Analysis

Apoptosis was measured based on the phospholipid phosphatidylserine translocation from the inner to the outer leaflet of the plasma membrane, which can be detected through annexin V staining. Necrosis is the process occurring in cells that lack plasma membrane integrity and could be stained by propidium iodide. OEC-M1 cells were cultured in a 6-well plate at a density of 150,000 cells per well. The cells were harvested 24 h after they had been treated with ZVI@mSiO_2_ NPs (10 μg/mL) alone or in combination with NH_4_Cl (20 mM) or vitamin C (200 μM). The harvested cells were stained with annexin V and propidium iodide (Annexin V-FITC Apoptosis Detection Kit; BD) for 10 min, and cell death analysis was then conducted using flow cytometer and software (FACS Canto II; BD).

### 4.9. Anticancer Efficacy Evaluation In Vivo

OEC-M1 cells (5 × 10^6^ cells) in the log phase were subcutaneously injected into the dorsal flank of NOD/SCID mice. The tumor sizes and body weights of tumor-bearing mice were measured twice a week until the tumor volume reached 100 mm^3^. Then, the mice were intravenously injected with a single dose of PBS or 100 μL PBS containing 1 mg ZVI@mSiO_2_ NPs (40 mg/kg). Three animals were used in each group. The tumor volumes were measured twice a week for 3 weeks. The tumor volumes were calculated using the following formula:

Tumor volume = 1/2 × (short axis)^2^ × long axis

All the animal studies were performed as per the protocol reviewed and approved by the Institutional Animal Care and Use Committee of National Cheng Kung University (IACUC No. 103279).

### 4.10. Statistical Analysis

All represented data are expressed as mean ± standard error. Statistical differences were evaluated using the Student’s *t*-test. The results were considered statistically significant at the 95% confidence interval (i.e., *P* < 0.05), but we provided all the *P* values. All figures shown in this article were obtained from at least three independent experiments (i.e., full replicates).

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
