# Peer review of "Iron Release Profile of Silica-Modified Zero-Valent Iron NPs and Their Implication in Cancer Therapy"

_ijms, 2019, doi:10.3390/ijms20184336_

Round 1

Reviewer 1 Report

The manuscript investigates the potential in anticancer-therapy of two different silica-modified zero-valent iron nanoparticles (ZVI NPs), evidencing that mesoporous-silica coated ZVI NPs are more active in inhibiting cancer growth in vitro than sealed silica ones. This different effect seems to be due to the different intracellular processes undergo by the two types of NPs: NPs with sealed silica shell are resistant to acidic lysosomal environment, this impeding the iron ion release, the ROS production and the consequent apoptosis induction; on ten contrary, mesoporous-silica coated ZVI NPs are sensitive to lysosomal environment, this leading to a massive ROS formation and death for apoptosis.

The in vivo experiments have been carried out only with mesoporous-silica coated ZVI NPs.

The topic of the research is interesting, but the manuscript suffers of some criticisms.

1) the Authors define the type of observed cell death as apoptosis, but a deeper analysis of the phenomenon is required with the aim of distinguish among apoptosis, ferroptosis or necrosis.

In this view, the following parameters have to be evaluated: release of lactate dehydrogenases, lipid peroxidation, caspase activation, DNA fragmentation, intracellular antioxidants). The evaluation of NP effect has to be performed also in presence of iron chelating molecules.

Moreover, the evaluation of cell viability by flow cytometry has to be performed also in early times (4, 8 or 12 hours) in order to verify if the high percentage of both annexin- and propidium iodide- positive cells are in secondary necrosis after having undergoing apoptosis.

2) to directly confirm the involvement of lysosomes in determining iron ion release the integrity of organelles need to be evaluated (for example, Acridin Orange, Lyso Tracker, or Galectine-3).

3) the Authors affirm that the evaluation of intracellular iron release has been carried out cytosolic fraction, but this is not correct: they centrifuged lysed cells at 20000 xg for 20 minutes (400,000 g total), whereas the cytosolic fraction is obtained after administering 6X106 g). The used fraction also contains endoplasmic reticulum (microsomes) and heavy mitochondria.

4) the authors have to discuss why in vivo experiments have been performed only with ZVI@mSiO2

Minor criticisms

1) Material and Methods

4.3: does the culture medium contain no antibiotics/antimicotics?

4.4: the number of cells seeded and the amount of NPs used have to be added.

4.6 and 4.7: why a different number of cells has been seeded for the different analyses?

In 4.7 "co-cultured" has to be changed since this world refers to the contemporary presence of different types of cells.

4.8: the statement "apoptotic cells lose membrane integrity" is incorrect, because the loss of membrane integrity is a feature of necrosis. In early apoptosis only changes in lipid distribution occurs, without loss of integrity.

4.9: the amount of ZVI@mSiO2 injected (1 mg) was the same for all animals or it was related to the weight?

4.10: the Authors affirm that both Student's test and AVOVA have been performed, but in the legends of the Figures only Student's test is reported.

Moreover, information about the results of statistical analysis (* or letters) has to be added also in the figures, not only in the legends.

The manuscript needs a careful revision to eliminate some mistakes.

Reviewer 2 Report

The paper presented herein, written by Yang et al., described the development of zero-valent iron nanoparticles (NPs) for cancer therapy. The aim was to show that their nanoparticles are suitable for a rapid release of iron inside the cell to increase ROS production and trigger apoptosis. 

Their particles are prepared with a mesoporous silica coating to insure better cell biocompatibility. They use ZVI NPs coated with plain silica as a negative control. The characterization of the particle is limited to TEM imaging and EDS analysis. Somehow there is a lack of characterization of the silica coating in order to validate the approach. For example, an analysis of porosity would have been beneficial. The iron content is not mentioned. As the internalization of NPs in cell is different between both NP types the overall quantity of iron is different. But their way to calculate iron concentration in cell and differentiate the part coming from intact particle solve this problem. 

Toxicity of ZVI@mSiO2 is well done. We could deplore the lack of a control cell line (non-cancerous) as the authors argue it will not have the same effect.

If I do follow very well the experimental setting and the conclusions on the cell activity of ZVI@mSiO2, I am not happy with the conclusion of the paragraph 2.3. Indeed, authors argue that there is a change in conformation of the NPs that become hollow. They compare a TEM Image between ZVI@SiO2 and ZVI@mSiO2. This figure 3 is note convincing to me. Indeed, we cannot honestly see a difference. Better images could be of great helps. Otherwise I think there is a bit of overinterpretation from these data. The same in the discussion part in which this matter is discussed. I would advise, if the imaging of the particles in cell is difficult, to use the same protocol as the one of iron content calculation to harvest the mesoporous silica particles after iron solubilization and perform high resolution TEM. 

The in vivo results are quite impressive considering there is only one intravenous injection of particle. An histologic study of the tumor would add some value to the paper. Specially to find out is we can find a lot of apoptotic cell. But I think it is not mandatory. However, in the discussion the quantity of iron delivered to achieve this goal should be correlate to the threshold concentration of iron toxicity. 

The scheme 1 is nicely done and try to summon the results of the paper. I do think it is a bit misleading. Indeed, the cell membrane are impermeable to any ions. An ion can go through a cell membrane only at two conditions: a mechanical or chemical breakage of the membrane or the presence of ion transporter. None of this hypothesis have been discussed in the paper. The scheme should be therefore modified in consequence. 

Some minor change could be done on the materials and methods section. 

First on the cell type use. I would expect a description of the phenotype (epidermal cell, mesenchymal cell …). 

In the in vivo data, the number of mice use is never mentioned. 

Line 135 there is a typo, change “ZVI@SiO2 or ZVI@SiO2” by “ZVI@SiO2 or ZVI@mSiO2”

Overall I think this is a nice piece of work that only need minor change to be suitable for publication. 

Those modifications are : 

-      Correct the typo mentioned above

-      Add the informations on the cell type use and on the number of animals used

-      Find better images to prove the conformation change of the particle in fig 3. 

-      Modify the scheme 1 and discuss how iron can be transfer from endolysosomal compartment ton cytosol

-      Discuss on the in vivo results according to the iron concentration injected. 
